# Privacy-Preserving Human Action Recognition with a Many-Objective Evolutionary Algorithm

**DOI:** 10.3390/s22030764

**Published:** 2022-01-20

**Authors:** Pau Climent-Pérez, Francisco Florez-Revuelta

**Affiliations:** Department of Computing Technology, University of Alicante, 03690 Alicante, Spain; francisco.florez@ua.es

**Keywords:** wrist-worn accelerometer data, human action recognition, evolutionary algorithms, wearable sensors, activities of daily living, de-identification, privacy-by-design

## Abstract

Wrist-worn devices equipped with accelerometers constitute a non-intrusive way to achieve active and assisted living (AAL) goals, such as automatic journaling for self-reflection, i.e., lifelogging, as well as to provide other services, such as general health and wellbeing monitoring, personal autonomy assessment, among others. Human action recognition (HAR), and in particular, the recognition of activities of daily living (ADLs), can be used for these types of assessment or journaling. In this paper, a many-objective evolutionary algorithm (MaOEA) is used in order to maximise action recognition from individuals while concealing (minimising recognition of) gender and age. To validate the proposed method, the PAAL accelerometer signal ADL dataset (v2.0) is used, which includes data from 52 participants (26 men and 26 women) and 24 activity class labels. The results show a drop in gender and age recognition to 58% (from 89%, a 31% drop), and to 39% (from 83%, a 44% drop), respectively; while action recognition stays closer to the initial value of 68% (from: 87%, i.e., 19% down).

## 1. Introduction

Population pyramid graphs can tell a lot about a nation and its future challenges. In terms of demographics, many developed countries show what is called a *reversed* or *constrictive* population pyramid, with large numbers of older individuals and a much smaller number of younger people. This entails a series of challenges to social and wellbeing systems in these countries since healthcare providers (often state-owned and publicly funded) require ever larger amounts of resources to cope with the increasing demand of services for older and frail people, according to Rashidi and Mihailidis [1]. All this, while the active workforce shrinks due to smaller generations of younger people (low natality rates), combined with an insufficient well-established flow of immigration, which could bring larger revenues to governments via income tax, which could translate into more funding for such care services.

It is in this context that active and assisted living (AAL) technologies and services can alleviate some issues caused by these demographic changes, as stated by Calvaresi et al. [2], since AAL aims at improving the quality of life of older people, enabling them to live autonomously for longer, in their homes or care facilities. All this while their families, caregivers and administrations in charge can be assured of their wellbeing; therefore, minimising the need for more intrusive, expensive, and labour-intensive means of care.

Within AAL, human action recognition (HAR), and specifically the recognition of activities of daily living (ADLs), is a good means of assessment of general health and wellbeing (i.e., answering the question “is the user carrying out their normal activities at the usual times?”); as well as useful for automatic journaling, or *lifelogging* [3]; that is, to perform self-reflection or remembrance of the activities carried out during the day, to help with the earlier symptoms of dementia (answering: “what did I do today?”). An automated system, collecting information of this kind during longer periods of time (e.g., that proposed by Munstermann et al. [4]), could also be useful for long-term autonomy level assessment, given that lowered activity performance scores or forgetting to carry out some routine activities altogether, could be indicative of mental decline, and could be helpful to further assess an end-user of such system.

This, of course, brings many ethical conundrums since it involves very sensitive user data and many decisions to be made by the users, or their next of kin, should the user no longer be capable of making these arrangements. Privacy is, therefore, paramount and needs to be considered from inception, following privacy-by-design and privacy-by-default paradigms, which are part of the data protection mandates of the European Union (i.e., the GDPR). The identity of the users should be protected, especially when data from different users is aggregated together in a database in such a way that sensitive aspects of the life of the users could be accidentally revealed.

In the context of mass collection and remote, centralised analysis of personal data, it is not possible to safeguard the use of the data as intended by the user. Collected data can be leaked, and the user has to trust service providers to properly protect their data. The GDPR and similar regulations require that the purpose of the data collection be clear and that, should the purposes change, consent from the user for these other purposes should be requested. By guaranteeing that the data itself hinders the usage for other purposes (as the proposed method intends), end-users can be reassured that it is not possible to use their data to recognise or identify them, while the services that they have requested (e.g., activity monitoring) can still be provided.

## 2. Motivation

Within AAL, some solutions entail the use of computer vision algorithms, using cameras installed in the environment, but these are often faced with rejection on the part of end-users or their families and caregivers. Another option is to use less intrusive means of user monitoring, such as environment-deployed sensors (e.g., pressure mats, door/cabinet opening sensors, etc.) or accelerometer and inertial magnetic unit (IMU)-based solutions. The former can be sometimes limited regarding the amount of information that can be collected about the users, whereas the latter is more suitable to understand the types of activities that the user performs.

Several such devices exist in the literature: some are worn mounted on the chest, or several IMU units are worn in different limbs [5,6] (e.g., one around the forearm of the dominant hand, another around the ankle, and so on). However, later devices use a single accelerometer embedded in a bracelet, such as many common smartwatches and sports monitoring bands. The Empatica E4 [7] is one such device that has been classified as a medical-grade device by several certification bodies in Europe and America. Apart from acceleration data, this device offers other capabilities too, to monitor dermal conductivity, skin temperature, heart rate, blood volume pulse, etc.

Human action recognition, and ADL recognition particularly, is an active field of research [6,8,9,10,11]. For instance, Gomaa et al. [8] take data from a smartwatch, compute the autocorrelation function up to a certain lag, then feed these features to a ‘random forest’-based classifier for training. For their experiments, they use a dataset with 14 different class labels, consisting of a mixture of motion primitives (e.g., walk, lie in bed, stand from sitting), as well as more complex activities (ADLs). A different, deep learning-based approach is that taken by Lu and Tong [9], in which they encode 3-axis signals as 3-channel images using a modified recurrence plot (RP) and train a tiny residual neural network to perform image classification. Their results are provided in the ADL dataset [12], as well as their own, the ASTRI dataset, which contains five different class labels (consisting of motion primitives only), performed by 11 subjects. Chen et al. [10] also use neural networks, in their case, ensemble stacked auto-encoders (ESAEs) with one-class classification based on the convex hull (OCCCH) for fall detection (five different fall types), and compare them to normal activities. Their dataset consists of four to five types of falls (slipping, tripping, etc.), which can happen frontally, backwards, or sideways (totalling 13 different fall events), as well as 16 normal activities (a combination of motion primitives with some ADLs). Jalal et al. [6] propose a manual feature extraction process, followed by feature selection and classification using a genetic algorithm (GA). They validate their method on a novel dataset (named IM-WSHA, with 10 subjects and 11 ADLs), as well as on two well-known datasets for comparison (WISDM [13] and IM-SB [14]). Finally, Fioretti et al. [11] extract a series of features from the accelerometer data from wrist-worn devices and select the best candidate machine learning algorithm from a series of six classifiers: decision trees (DT, J48 algorithm), random forest (RF), naïve Bayes (NB), neural networks (NNs), k-nearest neighbour (kNN), and support vector machines (SVMs). The J48 and RF classifiers performed the best, with recognition accuracies higher than 98% on their dataset consisting of 6 ADLs (pertaining to hygiene and house cleaning scenarios), performed by 36 participants.

However, none of the above methods was designed with privacy in mind, even when it has been shown [15,16,17] that accelerometer data can reveal the personal identity by the way each individual moves (i.e., reveal their gender, age, and other identity *markers*), and has to be considered a *quasi-identifier* [15]. To address this issue, Boutet et al. [18] present a way to *sanitise* motion data from accelerometers using generative adversarial networks (GANs) in a way that action recognition accuracy is maintained (3% difference), i.e., keeping data utility, while heavily (41% drop) reducing gender recognition.

Based on the biological concept of evolution, evolutionary algorithms (EAs) are aimed at finding the *fittest* ‘individual’ in a population by encoding a series of ‘traits’ of each individual in a ‘genetic code’ (therefore, often also called *genetic* algorithms, GAs). A good compilation of the latest algorithms, trends, and applications of these is presented in the review by Katoch et al. [19]. This code is often represented as a 1D vector $\vec{v}$ of *n* elements that can take binary values ($\vec{v}\in {\left\{0,1\right\}}^{n}$) or be real-valued ($\vec{v}\in R^{n}$). These values can encode parameters for the training of a machine learning (ML) classifier. This situation, in which the fitness function is the classifier that is being trained (also called the *inductor*), and calculating the fitness consists of training it with the values from $\vec{v}$, is called a *wrapper* approach, as explained by John et al. [20]. During the execution of the EA, a population is kept, and the fit of each individual is evaluated. After ranking, unfit individuals are discarded (the lowest ranks), and the fittest individuals are able to reproduce, thus keeping the population stable. A crossover function is used for this purpose. Furthermore, apart from single objective EAs, multi-objective evolutionary algorithms (MOEAs) have more than one objective (exactly two) to minimise, whereas many-objective (MaOEAs) can minimise several (more than two) objectives at once.

In a previous work, Poli et al. [21] propose the use of multi-objective evolutionary algorithms (MOEA) to minimise gender recognition while maintaining the ability to perform action recognition. The MOEA tries to find appropriate weights for the different features extracted from the accelerometer data before proceeding with the recognition. As a continuation of that work, in this paper, the aim is to introduce more traits to de-identify the person performing the activities by adding a second objective to minimise: not only gender recognition but also age. It is, therefore, required to consider a many-objective evolutionary algorithm with three objectives: maximising the recognition of the action being performed and minimising the user’s gender and age recognition. Another goal of the paper is to validate the approach using a larger dataset than those previously available. The remainder of this paper is organised as follows: Section 3 will introduce the pre-processing of the data for gender and age recognition minimisation and action recognition maximisation, as well as the many-objective evolutionary algorithm (MaOEA) used. Section 3.4 will introduce the experimental setup used to validate the proposed approach. The results are then introduced in Section 4. Finally, there is a discussion and conclusions in Section 5.

## 3. Materials and Methods

### 3.1. Dataset and Data Pre-Processing

To validate the proposed approach, the publicly available PAAL ADL Accelerometry dataset v2.0 [22,23] is used. This has been collected as part of the efforts by the authors to create a large dataset with data from 52 participants (26 men, 26 women), aged between 18 and 77 years old, performing 24 activities of daily living, including some motion primitives. Participants wore the Empatica E4 bracelet around the wrist of their dominant hand during the experiments. This data-capture wristband is equipped with an accelerometer that is configured with a range of ±2 g and a resolution of 0.015 g, and has a sampling frequency of 15 Hz [23]. Furthermore, there is no formal set of instructions provided to the participants, other than to perform activities independently (e.g., assuming each repetition is independent of each other, as an example: the participant should assume their hands are dry each time they wash them, even if they just washed them for the previous repetition). As depicted in Figure 1, each individual (IDs on the *x*-axis) provided an average of 5 repetitions for each of the 24 activities (*y*-axis labels), with some cases of up to 7 repetitions; totalling 6072 sequence files. Additionally, Figure 2 shows the distribution of participants by gender and age group.

Following the steps of previous works [11,21], features are extracted both from the time and frequency domains. First, a fourth order Butterworth low-pass filter with a cut-off frequency of 15 Hz is applied to eliminate the high-frequency noise, but preserve human motion and gestures (human motions are very seldom above 15 Hz). After that, a third order median filter is applied to remove abnormal spikes in the time-series data.

In this work, the time domain features proposed by Fioretti et al. [11] were extracted. These features consist of the: mean, median, σ, max and min, range, axes correlation, signal magnitude area, coefficient of variation, median absolute deviation (MAD), skewness and kurtosis, autocorrelation, percentiles (20th, 50th, 80th, and 90th), interquartile range, number of peaks, peak-to-peak amplitude, energy, and root-mean-square (RMS). In addition to that, Poli et al. [21] propose to calculate some further frequency-domain features, namely: spectral entropy, energy, and centroid; mean, and σ, as well as percentiles (25th, 50th, and 75th). Some of these features are extracted separately for each axis (*X*, *Y*, and *Z*) or for the signal magnitude vector (SMV) of each sample. The time-domain features were extracted based only on the changes in the acceleration, whereas the frequency domain analysis entailed computing the magnitude of the discrete fast Fourier transform (FFT) on the acceleration data. The 62 resulting features are shown in Table 1.

To obtain the samples from the 6072 sequences, the data in each sequence is split into windowed segments [24]. Therefore, features are extracted from the data values of each fixed-size sliding window of 5 s, or 160 samples, at 32 Hz, which is the capture rate of the Empatica E4 accelerometer sensor [7]. Furthermore, the data sequences are split in such a way as to have a 20% overlap between adjacent windows. This leads to a total of 28,643 data samples. The code to split the raw data, as well as to calculate the features for each sample, is available online [25].

### 3.2. The *Inductor* Classifier: Random Forest

Among the many available classifiers for the task, Random Forest (RF) is chosen due to its good performance, as reported by previous studies [11,26]. During training, RF constructs multiple trees (therefore a *forest*), or *estimators*, where each tree is grown with a subset of features picked at random. The whole set of features is used to train the random forest. Poli et al. [21] conducted a study to determine the number of trees needed to achieve the best performance without incurring unnecessary additional resource consumption and set the optimum number of trees to be 90. The model is trained using *k*-fold cross-validation, in which the data is split into *k* subsets (folds or splits). The value of *k* was set to 10; that is, 10 different models are trained, using k−1=9 folds for training, and the remaining one is used for testing. The final result is then obtained as the average of all *k* tests.

### 3.3. The Many-Objective Evolutionary Algorithm: NSGA-III

The nondominated sorting genetic algorithm III (NSGA-III) [27,28] is a many-objective evolutionary algorithm based off of a previous version called NSGA-II [29], which was only multi-objective (i.e., 2-objective). However, NSGA-III is able to perform many-objective optimisations from 3 to up to 15 objectives. In order to guide the exploration of the Pareto-optimal front, the authors introduce a reference point-based framework that emphasises population members that are nondominated yet close to a set of supplied reference points. These reference points are distributed along a series of reference *directions* or partitions (as explained in: https://pymoo.org/misc/reference_directions.html, accessed on 19 January 2021), denoted as *p*. The number of resulting reference points *n* is determined by:(1)$$n=C_{p}^{M+p-1}\;\;,$$
where *M* is the number of objectives to optimise. Since the population size is set to 50 in the experiments, and there are M=3 objectives (age, gender, and action), *p* is set to 8 so that it results in:(2)$$n=C_{8}^{3+8-1}=C_{8}^{10}=\left(\begin{array}{c} 10\\ 8 \end{array}\right)=45\;\;,$$
which is a number smaller than the chosen population size of 50 individuals. Any larger value of *p* would exceed this and result in unexpected behaviour by the algorithm.

Finally, for the choice of crossover and mutation, the simulated binary crossover (SBX) and polynomial mutation (PM) were used, respectively.

Similarly to [21], NSGA-III is employed for feature weighting, i.e., to find the appropriate weights, so that action recognition is maximised while gender and age recognition is minimised. Therefore, each individual is considered as a real vector in which each gene represents the weight given to each feature.

### 3.4. Experimental Setup

The total number of training processes to perform per generation is very large, and even more for the initialisation of the algorithm. That is because the size of the population (50 individuals) has to be used to train *k* folds (10 folds), with three different classifiers (gender, age, and action), leading to a total of 50×10×3= 1500 random forest training processes. For each subsequent generation, the number of processes is smaller since *only* 10 new individuals are added to the population, for which the *fitness* is unknown; therefore, leading to a total of 10×10×3=300 RF training processes.

Due to all this, a largely parallel multiprocessor machine is desirable. The Nvidia DGX A100 machine is aimed at general purpose GPU calculations (GPGPU), using CUDA-accelerated code (usually used for the training of deep neural networks). However, the machine is also equipped with 2 CPUs of 64 cores each and 2 hardware threads per core, totalling 256 threads/processes that can be launched in parallel. Making use of the ability of the Python interpreter to perform multiprocessing tasks using process Pools and the starmap() function from the multiprocessing library, 200 parallel processes are launched at once, each receiving a triplet of classifier, fold, and population individual (feature weights). This heavily accelerates training times for the RF algorithm, which in turn accelerates the evolutionary process of the employed NSGA-III many-objective algorithm.

The Random Forest (RF) classifier used as the *inductor* is the implementation provided in the *Scikit-learn* [30] Python library. The MaOEA NSGA-III algorithm is provided by the *pymoo* library [31].

## 4. Results

### 4.1. Baseline: Initial Individual

The baseline is obtained by considering all features as unmodified; that is, equivalent to multiplying each feature by 1.0. This baseline is used to determine the degree of optimisation achieved by the best individual of the MaOEA algorithm that is described next. The results of this initial experiment are shown in the first column of Table 2, and on the confusion matrices as shown in Figure 3, for gender and age; as well as Figure 4, for action (HAR); and will be discussed in greater detail there. The first column of Table 2 shows the results for the *initial* individual, i.e., the one in which all features are equally weighted to 1.0. In all cases (gender, age, and action), the resulting accuracy is above 80%. It is close to 89% for gender, 84% for age, and 87% for action recognition. It is, therefore, shown that the features used for training are revealing age and gender traits of individuals; therefore, leaking this information. The table also shows the expected values for a *random choice* ‘dummy’ classifier, i.e., one that would always respond with the same prediction: for gender, it would be expected that such a classifier has an accuracy of 50%; whereas for age that would be closer to 14.3%, given that there are 7 age bins ($\frac{1}{7}=0.142857\ldots$); in the case of action classes, that would be lower, closer to 4.2%, given that there are 24 possible class labels. Therefore, the objective of the optimisation problem is to achieve values closest to 0% in the *(B-R)* column for age and gender while maintaining the accuracy for action recognition (HAR).

Looking at the results for gender more in depth, Figure 3a shows that 91% of men and 87% of women are correctly classified, and the most confused class is women, which are mistaken for men in 13% of cases: a plausible explanation could be that there are more men in two of the three largest age groups by the number of participants (i.e., ‘21–30 y/o’ and ‘31–40 y/o’).

In the case of age recognition, Figure 3b shows a quite strong black diagonal, meaning for most individuals, age groups are correctly predicted. There is, however, a faint grey band in columns 2–4 (age group comprised between 21 and 50 years old). This is probably due to the fact that these three age groups are larger than the rest (as depicted in Figure 2), and therefore, when in doubt the classifier gives either of these as the predicted class label. The most confused group is the 61–70 y/o, which is confused mostly with that of 41–50 y/o individuals (i.e., the largest group).

Finally, regarding action classification, Figure 4 shows that the most confused classes are ‘open a bottle’ and ‘open a box’ (with each other), maybe because they involve similar positions of the hands (this is also the case in Poli et al. [21]); as well as ‘put on a shoe’ and ‘take off a shoe’ (they are *symmetrical* along the time axis); ‘put on glasses’ and ‘take off glasses’ (same); stand up and sit down (ditto); as well as ‘phone call’, confused with ‘put on glasses’, which makes sense, since both involve similar movement of the hand towards the side of the head; finally, also ‘sneeze/cough’ and ‘blow nose’ (with each other, as they are similar), and ‘sneeze/cough’ which gets confused with several classes, including ‘put on glasses’, among others. This latter confusion could be due to the lack of movement of the wrist in this action, should the user prefer not to bring their hand to the mouth for sneezing or coughing, or bringing the elbow as is recommended, or variations thereof.

### 4.2. Optimised Results: The Best Individual after NSGA-III

To depict the optimisation undergone by the *individuals* in the population of the evolutionary algorithm over the course of the generations, Figure 5 shows the final set of solutions (individuals) of the population, showing black crosses for each individual, and a green dot representing the best *fit* individual (i.e., minimising age and gender recognition, while keeping action recognition as unaltered as possible). The results corresponding to the green dot are also shown in the respective column “*best (B)*” on Table 2. Please note that this figure is a 2D representation of a 3D scatter, and that the two panels (left and right) depict two *planes* (2D views) of the 3D volume. The planes are provided as 2D for clarity of visualisation. In each panel, a red dot is also included, representing the *initial* individual (all features weighted to 1.0), which was used to obtain the baseline results. It can be observed (left panel) that while the gender accuracy dropped almost 0.3 (≈30%, 30.6 as shown in Table 2, “*Diff. (B-I)*” column), the HAR (action) accuracy dropped 0.2 (≈20%, 19.5 as per Table 2). On the right panel, the age vs. action graph is plotted in the same manner. Similarly, it can be observed that, while the age accuracy dropped by 0.4 (≈40%, 44.3 as shown in Table 2), the action (HAR) accuracy dropped by 0.2 (≈20%). An *artefact* of the 2D representation is shown on the left panel, as the green dot (the best individual), has some black-crossed individuals above it (i.e., with better HAR accuracy, while keeping gender recognition *as low as* the chosen solution). However, this is not *real*, as this is, after all, a 2D view of the 3D volume, and it could well be that the crosses depicted above the green dot do not have an equally ‘improved’ correspondence on the right panel (i.e., keeping age prediction low, too).

From the best solution (green dot) in Figure 5, it is also worth noting the *contributions*, i.e., the weights, given to each of the 62 features extracted, as this is indicative of the importance of each of these regarding the leakage of information related to age and gender traits, or conversely, their importance for action recognition. Figure 6 shows this. The *x*-axis represents each feature, whereas the height of the bars represents the weight assigned to each feature at the end of the evolutionary algorithm for the best candidate individual. Due to the large variation in weight values, the *y*-axis is logarithmic in scale. It is worth observing the features that have almost been completely zeroed, as those would be the ones that reveal the most sensitive information, namely these are: the ‘Max SMV’, some ‘Range’ values, the ‘Interquartile range’, the ‘Energy’, as well as some frequency domain features, such as ‘Spectral entropy’ and ’Spectral energy’, as well as the ‘freq. Mean’ in most axes. The specifics as for why each has been excluded would lead to a long analysis, but it seems quite logical that frequency domain features are excluded, as these can contain many more information about some *harmonics* of motion frequency that are not only relevant for motion analysis, but also for particular individual activity performance; thus, leaking information about their identifying traits.

Regarding gender recognition, Figure 7a shows the confusion matrix of gender prediction vs. expected label. It can be seen that now only 18% of women are correctly classified, whereas most other samples are classified as belonging to men; thus, correctly achieving gender de-identification.

In the case of age recognition, Figure 7b shows the confusion matrix of age classification. In this case, the resulting feature weights also correctly allow age de-identification, since most samples fall in only two age groups (‘21–30’ and ‘31–40’). Most likely, the reason most samples fall into these categories is due to the larger proportion of participants in these two age groups.

Finally, Figure 8 shows the confusion matrix for ADL recognition (HAR) for the *best* individual of the population. It can be observed that certain classes are much worse affected by the optimisation than others. Such is the case of ‘take off a shoe’, which is confused with ‘writing’ and ‘washing dishes’, which initially seem very different activities from the point of view of wrist motions. However, some features have been excluded or greatly diminished in their contribution to the overall classification. Since some features that were calculated separately for the three axes may have been affected by the optimisation *individually*, it is possible that some class labels are now confused with some others that are quite different, only because they share similar values for the unaltered features.

## 5. Discussion

This paper has introduced the usage of a many-objective evolutionary algorithm for the purpose of finding the best feature set that performs human action recognition (ADL recognition) at an acceptable accuracy level (i.e., preserving data utility) while concealing identifying traits of users (gender and age). The drops in age and gender recognition are more than double (44.3 vs. 19.5) and 1.5 times (30.6 vs. 19.5) greater than that of ADL recognition. Gender recognition is quite close to random classification (minimum theoretical accuracy value), which is desired. Age classification has suffered the biggest drop (44.3%).

When compared to the work of Poli et al. [21], it can be seen that the drop in human action recognition of the proposed method (many-objective: age, gender, and action) is greater than that of the multi-objective presented in [21] (gender and action, only). However, it is worth noting that a many-objective problem, having an additional dimension to explore in the solution space, creates a much more complex problem to solve. This is so because solutions that could be optimal for two of the dimensions (i.e., gender and action), might be suboptimal for the other pair of dimensions (i.e., age and action). This has been shown in Figure 5, and discussed earlier.

Furthermore, it would be worth exploring the *symmetrical* nature of some activities along the time axis (e.g., ‘sit down’ vs. ‘stand up’, or ‘put on a shoe’ and ‘take off a shoe’). Since the model does not take any feature that is reflective of when (i.e., where in the sequence) a sample was taken from. This could be an improvement to be considered for future work.

Finally, since some extracted features are calculated individually for each axis (e.g., mean on *x*, mean on *y*, or mean on *z*), it could be worth exploring the idea of having a single real-valued variable (i.e., a *gene* in the code) modify all three axes at the same time. It could be possible that, it is not the axis of each individual feature, but the type of feature (e.g., mean vs. entropy) that is important to convey age, gender, or activity information, and thus, it would be possible to have fewer genes (i.e., real-valued variables) in each individual in the population of the evolutionary algorithm. This would, in turn, reduce the dimensionality and might facilitate convergence. It is possible that ADL recognition power might be preserved, while gender and age should not be affected as much since it would be logical to think that similar features in different axes should convey the same amount of *revealing* information. This is left for future work.

To conclude, this paper is part of an ongoing effort to improve user privacy in ADL recognition, and more broadly, in AAL technologies. Methods for ADL recognition in this context are much needed and are, and will be, released in future years and decades. It is important, however, to allot research efforts towards privacy-aware methods for recognition in order to improve user acceptance and social and personal wellbeing as a consequence.

## Figures and Tables

**Figure 1 sensors-22-00764-f001:**
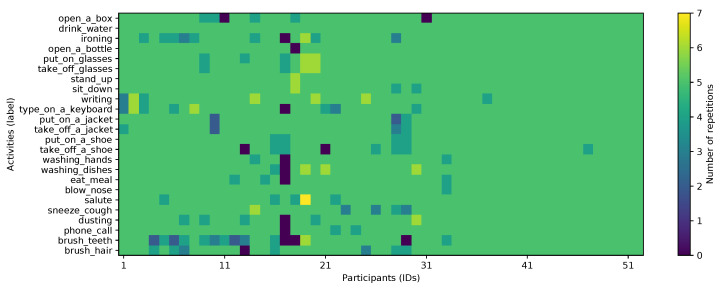
A matrix plot showing a summary of the collected dataset [22] used in this paper, showing number of instances recorded per participant and activity (reproduced from [23]).

**Figure 2 sensors-22-00764-f002:**
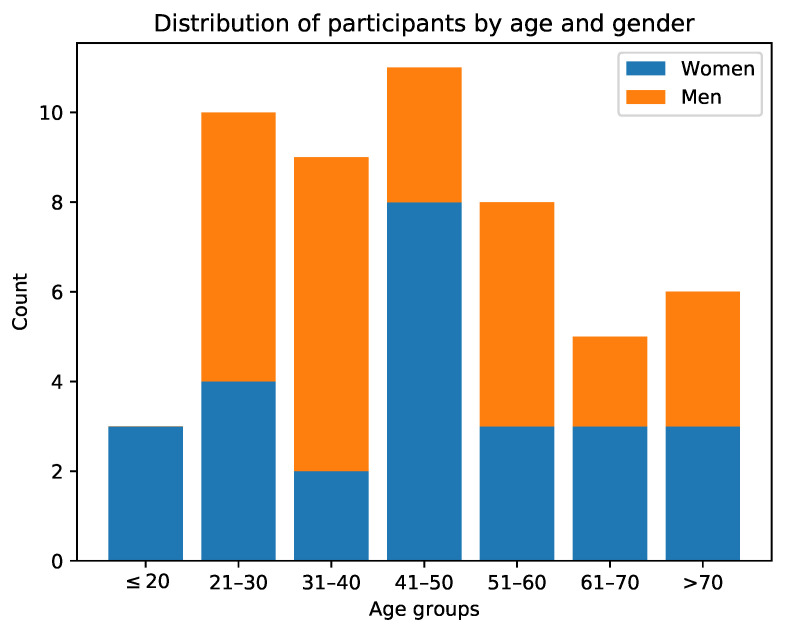
Histogram showing the distribution of participants among different age groups and genders (reproduced from [23]).

**Figure 3 sensors-22-00764-f003:**
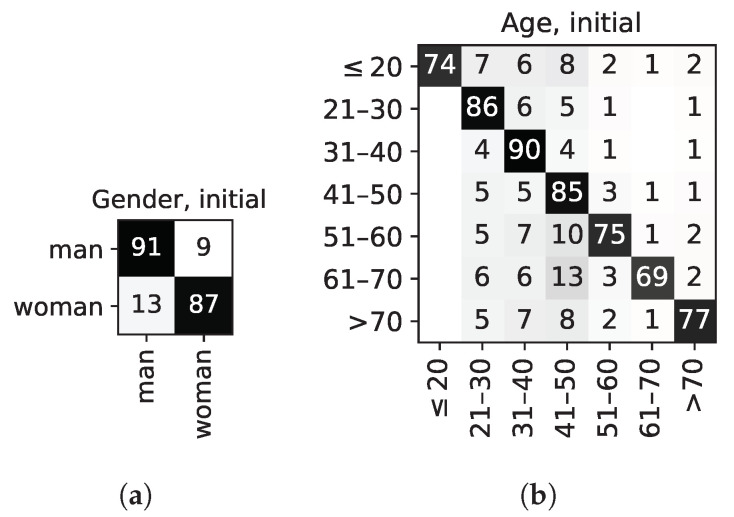
Confusion matrices for gender and age recognition, showing the results for the *initial* individual. It can be observed that the features used are quite good at revealing undesired, sensitive identity traits. Values are given as a % of the total number of samples per category. (**a**) Gender classification, initial individual. (**b**) Age classification, initial individual.

**Figure 4 sensors-22-00764-f004:**
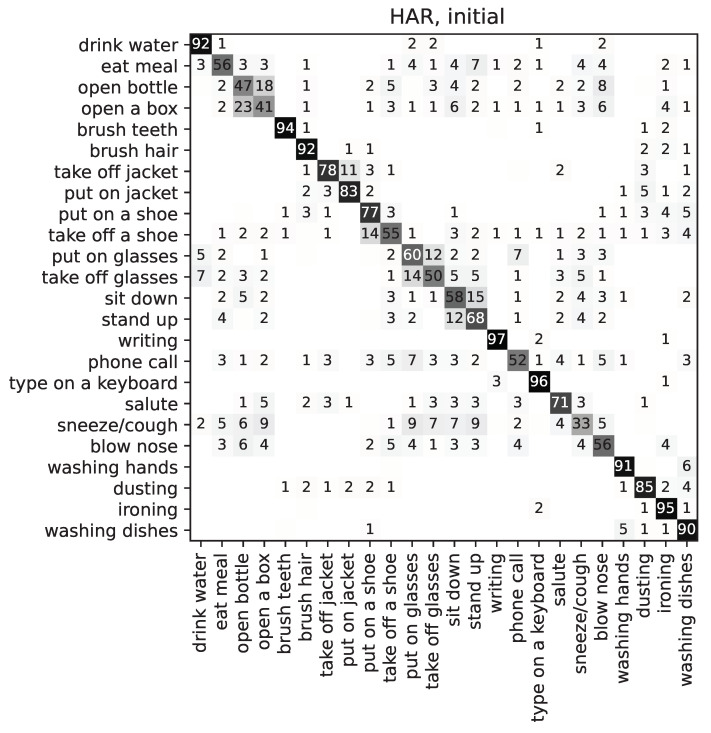
Confusion matrix for human action recognition, showing the *initial* individual (all features equally weighted to 1.0). Values are given as a % of the total number of samples per category.

**Figure 5 sensors-22-00764-f005:**
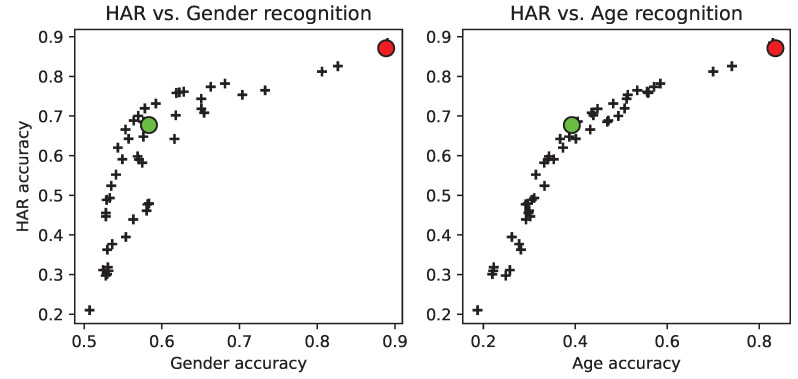
Final set of solutions (individuals) after running the NSGA-III MOEA algorithm. The best-performing (overall) solution is shown in green. The initial individual (all features equally weighted to 1.0) is shown in red. Black crosses represent individuals of the population in the latest generation. Gender accuracy range is in the range [0.5,1] (two bins), whereas age accuracy is $\in [\frac{1}{7},1]$ (seven bins).

**Figure 6 sensors-22-00764-f006:**
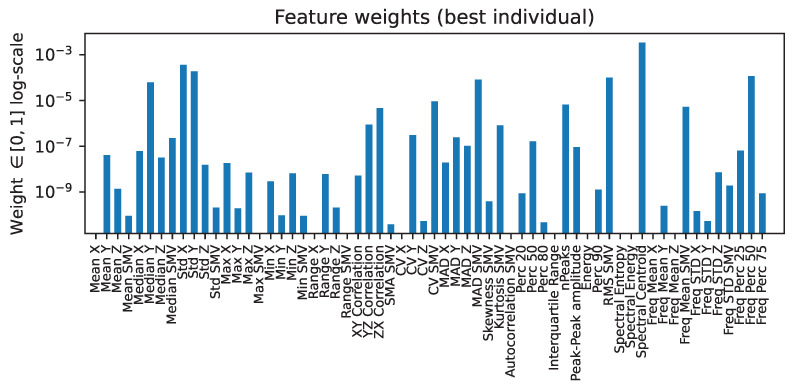
Contributions (weights) assigned to each feature in the best-performing individual (green dot on Figure 5).

**Figure 7 sensors-22-00764-f007:**
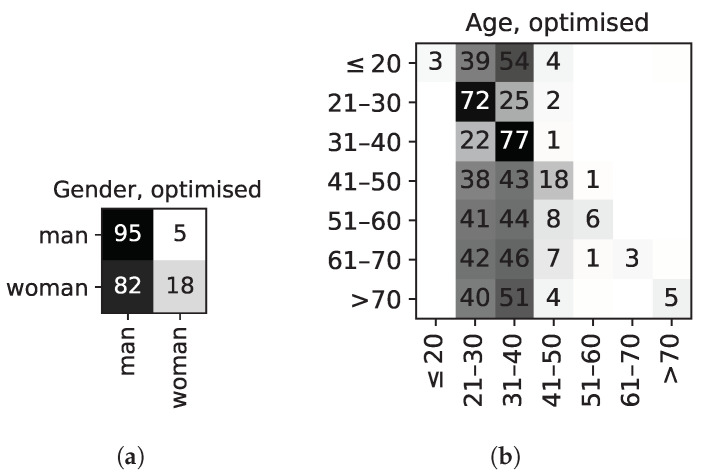
Confusion matrices for gender and age recognition, showing the results for the *best* (i.e., after optimisation) individual. It can be observed that optimisation obfuscates age and gender recognition, as desired. Values are given as a % of the total number of samples per category. (**a**) Gender classification, best individual. (**b**) Age classification, best individual.

**Figure 8 sensors-22-00764-f008:**
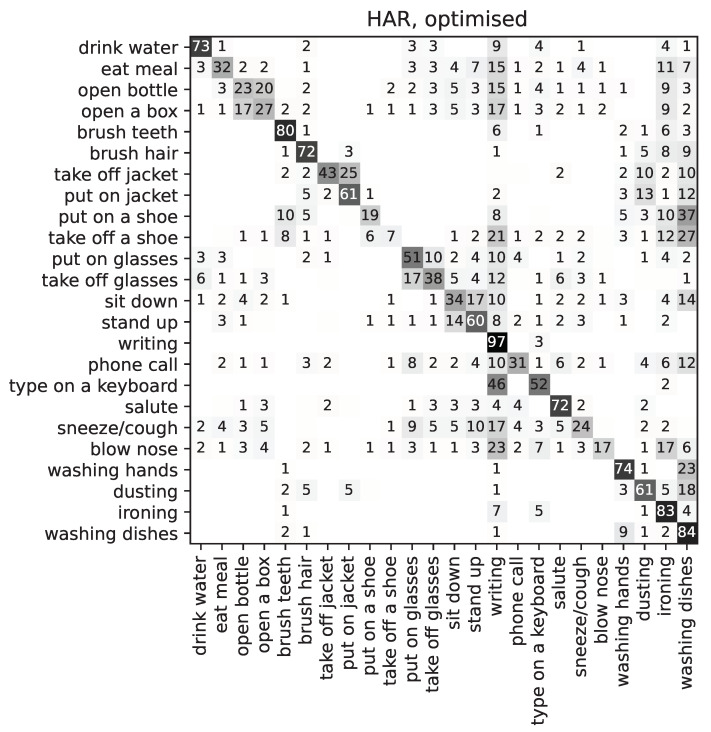
Confusion matrix for human action recognition, showing the *best* individual after optimisation (features weighted as per Figure 6). Values are given as a % of the total number of samples per category.

**Table 1 sensors-22-00764-t001:** Features extracted from the raw data.

Domain	Features	Computation
Time	Mean, median, σ	X, Y, Z; SMV
	Maximum, minimum, and range	X, Y, Z; SMV
	Correlation between axes	XY, YZ, ZX
	Signal magnitude area (SMA)	SMV
	Coefficient of variation (CV)	X, Y, Z; SMV
	Median absolute deviation (MAD)	X, Y, Z; SMV
	Skewness, Kurtosis, Autocorrelation	SMV
	Percentiles (20; 50; 80; 90), interquartile range	SMV
	Number of peaks, peak-to-peak amplitude	SMV
	Energy, Root mean square (RMS)	SMV
Frequency	Spectral entropy, energy, and centroid	SMV
	Mean, σ	X, Y, Z; SMV
	Percentiles (25; 50; 75)	SMV

**Table 2 sensors-22-00764-t002:** Results as overall accuracy (Acc. %) for the initial individual (I) in which all features are equally weighted by 1.0, as well as the best individual after optimisation (B), including the difference (%), as well as the expected performance (%) of a random choice classifier (R).

Classifier	Initial (I) Acc.	Best (B) Acc.	Diff. (B-I)	Random	Diff. (B-R)
Gender	88.9	58.3	⇓ −30.6	50.0	8.3
Age	83.5	39.2	⇓ −44.3	14.3	24.9
HAR	87.2	67.7	↓ −19.5	4.2	—

## Data Availability

Not applicable.

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
