# Peer review of "Privacy-Preserving Human Action Recognition with a Many-Objective Evolutionary Algorithm"

_sensors, 2022, doi:10.3390/s22030764_

Round 1
Reviewer 1 Report
This paper discusses the possibility of maintaining privacy of identifying features such as gender and age from a dataset containing sensor data for activities of daily living. This research could have important consequences within the right use case.
My biggest concern with the presented research is that the authors seemed to have solved the problem of identifying differences in gender and age at the expense of accuracy of their classifier. Is it not the intention of the classifier to correctly detect differences in various ADL’s? Therefore, I’m confused as to why accuracy should be sacrificed to protect the gender and age identities of participants. Surely there are other data protection mechanisms that could be put in place to protect these. Perhaps the data management plan needs to be investigated in these circumstances to protect gender and age identifiers. And if a properly configured algorithm has not been set up to evaluate gender but instead just focus on ADL identification, then how can these traits even be identified in the first place?
Please address my comments to your paper.
The paper could be improved by proof-reading it in more detail and perhaps getting feedback from a native English speaker on the papers grammatical structure.
I’m not sure what you mean by the first sentence whereby you state that “Many developed countries show what is called a reversed demographic pyramid”. What do you mean by this statement? Please add some text to this sentence to clarify what you mean.
The authors state in line 60 that IMU’s are “…. worn in different limbs (e.g. one around the forearm of the dominant hand, another around the ankle”. Usually a smart device containing an IMU sensor is worn on the non-dominant hand so that any movement that it records is a better representation of the general activity of the wearer. For example, if the IMU sensor is worn on the dominant hand, additional actions such as pouring water from a kettle, or writing a document will be picked up by it. These actions will not be sensed on the non-dominant hand. Therefore I suggest that the authors should clarify this point and update their paper accordingly. And I also suggest that the authors should clarify whether the dominant or non-dominant hand was recorded within their datasets used in their study.
The authors used the PAAL ADL dataset v2. It would benefit the paper and the readers understanding of how this research could be used with various types of hardware if the hardware used to record the dataset is discussed in some detail. Please add some information regarding the types of sensors that are within the E4 wristband.
Furthermore, do the authors have any additional information on the processes that were used for the test environment used for this dataset? For example, how were each of the ADL tasks recorded? I’m assuming that each participant in the study had to complete each ADL by following a formal set of instructions. Please add some background information on the laboratory setup that was used for this study.
I’m not sure what thoughts the authors are attempting to convey in lines 343 – 347. Please restructure this section to make it easier to understand what thoughts the authors are attempting to convey to the reader.
Please explain in some detail why you think it is more important to protect the gender and age identity of data recorded from participants than it is to maintain the ability to accurately recognise ADL. If the core reason for a classifier is to identify ADL’s, then why is it important to hide participant identity if gender is not under investigation?
Reviewer 2 Report
The authors present the article entitled “Privacy-preserving Human Action Recognition with a Many-objective Evolutionary Algorithm”. The article is easy to read and is well structured. However, there are some concerns:
The manuscript is found with some grammatical and typographical errors. The authors are suggested to go through the manuscript thoroughly and get it to proofread for grammatical and typographical errors
Introduction section: References are missing. It is needed to support the research background.
Avoid using apostrophes.
Figure 1 caption: I recommend moving the following sentence to the main text: “It shows the number of repetitions per activity (y-axis labels) and participant (IDs, x-axis). As explained, participants provided, on average, 5 repetitions of each of the 24 activities considered, totaling 6,072 sequences (reproduced from [17]).”
In line 84, can you discuss the advantages of up-to-date GA already reported as:
Non-linear regression models with vibration amplitude optimization algorithms in a microturbine; Self-tuning neural network PID with dynamic response control; A new methodology for a retrofitted self-tuned controller with open-source fpga
I argue to the authors to present a flow chart to summarize the proposed evolutionary algorithm to be more attractive.
Figure 3 and 7: Identify sub-plot with a) and b) instead of right (line 248 and 308)
